# Score-Based Image-to-Image Brownian Bridge

## ABSTRACT

Image-to-image translation is defined as the process of learning a mapping between images from a source domain and images from a target domain. The probabilistic structure that maps a fixed initial state to a pinned terminal state through a standard Wiener process is a Brownian bridge. In this paper, we propose a score-based Stochastic Differential Equation (SDE) approach via the Brownian bridges, termed the Amenable Brownian Bridges (A-Bridges), to image-to-image translation tasks as an unconditional diffusion model. Our framework embraces a large family of Brownian bridge models, while the discretization of the linear A-Bridge exploits its advantage that provides the explicit solution in a closed form and thus facilitates the model training. Our model enables the accelerated sampling and has achieved record-breaking performance in sample quality and diversity on benchmark datasets following the guidance of its SDE structure.

## 1 INTRODUCTION

Analogous to automatic language translation, image-to-image translation is defined as the process of learning a mapping between images from a source domain and images from a target domain. The applications of image-to-image translation are diverse, spanning areas like image colorization [5], generating semantic labels from images [17], image super resolution [6, 18, 40], and domain adaptation [26].

Generative Adversarial Networks (GANs) [12] are a powerful framework for image-to-image translation, as they can learn a mapping function that preserves the content of the source image while generating realistic and diverse outputs. Additional information, such as class labels or input images, can be added to guide the generation process [39]. Despite their remarkable capabilities, GANs face challenges that warrant attention. First, they are notoriously difficult to train efficiently [3]. Second, GANs often suffer from the issue of dropping modes in the output distribution [25].

Diffusion models [15] have demonstrated competitive performance in generating high-quality images when compared to GAN-based models [8]. Unlike many conditional diffusion models that approach image-to-image translation as conditional image generation [30], the Brownian Bridge Diffusion Model (BBDM) [22] offers a unique perspective. BBDM treats image-to-image translation as a stochastic Brownian Bridge process, the probabilistic structure that maps a fixed initial state to a pinned terminal state through

a standard Wiener process, and directly learns the translation between two domains. This novel approach provides an alternative avenue for achieving effective and high-fidelity image-to-image translation.

Expanding the exploration of innovative approaches in image-to-image translation, score-based generative modeling through Stochastic Differential Equations (SDE) emerges as a compelling paradigm. The integration of SDE in generative modelling brings forth novel sampling procedures and enhances modeling capabilities, presenting a promising avenue for image generation [34]. In this paper, we propose a score-based SDE approach via the amenable Brownian bridges (A-Bridges, defined in Section 3), for image-to-image translation as an unconditional diffusion process. As shown in Figure 1, the A-Bridge formation streamlines the inference and generative processes for a large family of Brownian bridge models that take advantages of the amenable structure. Our linear A-bridge model (defined in Section 4) simultaneously addressed three key requirements in image-to-image applications including: high sample quality, diversity coverage, and fast sampling, and achieved record-breaking performance in sample quality and diversity when compared to the current state-of-the-art (SOTA) image-to-image translation methods. The main contributions of this work can be summarized as follows:

(1) We propose a score-based Brownian bridge scheme based on SDEs that sets a solid theoretic ground for image-to-image translations. The A-Bridge SDEs streamline the unconditional forward and reverse diffusion processes and thus proffer better understanding and realization of image-to-image mappings.

(2) Our framework encompasses a large family of Brownian bridges that offer explicit solutions in a closed form, and thereby facilitates the training of the diffusion model. Among a variety of potential A-Bridges, we implemented and discretized the linear A-Bridge model because of its simplicity and clarity. The accelerated sampling algorithm is also provided.

(3) Through extensive experiments on various benchmark datasets, our linear A-Bridge model, following the guidance of its SDE structure, exhibits a significant enhancement over current SOTA for image-to-image translation tasks.

## 2 BACKGROUND

### 2.1 Image-to-image translation

The seminal work of pix2pix [17] marked the inception of Image-to-Image translation tasks, demonstrated the capability to translate images from one domain to another through conditional adversarial networks. Wang et al. [39] further advanced the field by employing Conditional GANs to synthesize high-resolution, photo-realistic images from semantic label maps, showcasing the potential for detailed and context-aware image translation. CycleGAN [43], on the other hand, introduced a novel approach that addressed the

*ACM MM, 2024, Melbourne, Australia*

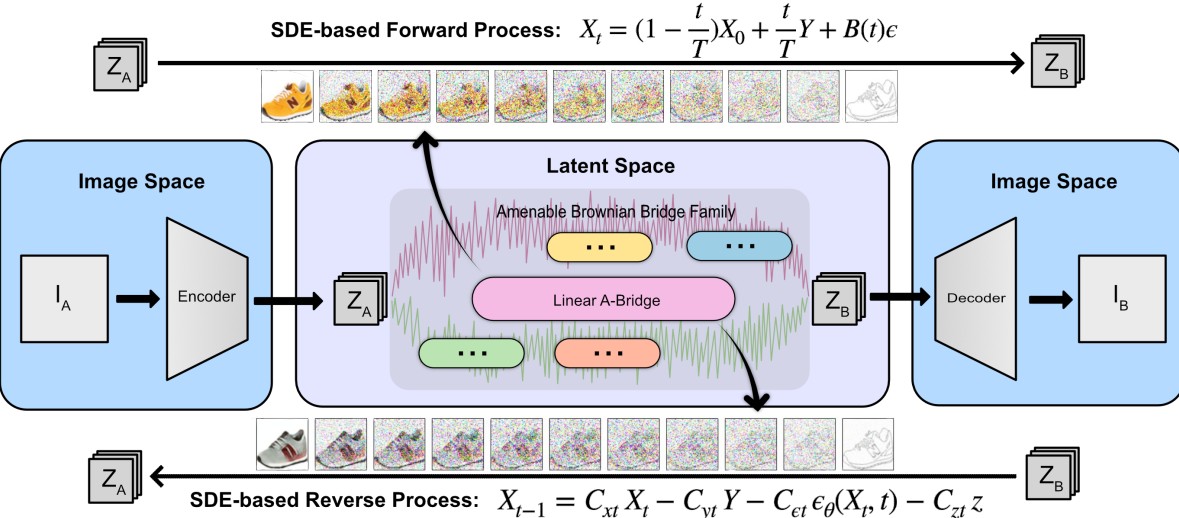

**Figure 1: Amenable Brownian Bridges (A-Bridges) for image-to-image translation. Images from domain A are transformed into a latent space representation via the VQGAN encoder [9], traverse through an SDE-based A-Bridge, and are then reconstructed in domain B using the VQGAN decoder.**

challenge of unpaired image translation, enabling the transformation of images from a source domain to a target domain without the need for paired examples. Building upon these foundations, DRIT++ [21] was proposed, which embeds images into domain-invariant content spaces and domain-specific attribute spaces, facilitating versatile image translation with a focus on shared information across domains. In a different vein, Han et al. introduced DCLGAN [13], leveraging contrastive learning within a dual learning setting to infer efficient mappings between unpaired data. Collectively, these contributions have significantly advanced the capabilities and scope of image-to-image translation techniques.

Recent advancements, exemplified by works like VQ-VAE-2 [9], leverage the strengths of Convolutional Neural Networks (CNNs) and transformers, combining the inductive bias of CNNs with the expressive power of transformers. This integration enables the modeling and synthesis of high-resolution images. However, it is noteworthy that GAN-based techniques, while effective, grapple with training instabilities and mode collapse issues [3, 25].

## 2.2 Diffusion Models

In the landscape of image-to-image translation, diffusion models have emerged as a compelling alternative to conventional GAN-based approaches. Notably, the diffusion probabilistic model (DDPM) [15] stands as a pioneering work, showcasing competitive performance for image synthesis. Recent advancements delve into leveraging DDPM within the latent space of powerful pre-trained autoencoders [30]. This distinctive approach enables the training of diffusion models on a representation that strikes a near-optimal balance between complexity reduction and detail preservation.

The integration of SDE [34] into generative modeling provides a fresh perspective. The reverse-time SDE, relying solely on the time-dependent gradient field or score of the perturbed data distribution, not only encapsulates but also extends previous score-based generative modeling and diffusion probabilistic modeling. This integration introduces novel sampling procedures and enhances modeling capabilities.

*2.2.1 Conditional Diffusion Models.* Conditional diffusion models [38] have been explored, treating image-to-image translation as a conditional image generation task. Another contribution comes from the Dual Diffusion Implicit Bridges (DDIB) method [35], which diverges from traditional training on domain pairs. DDIB employs a distinctive two-step process: it first acquires latent encodings for source images using the source diffusion model and subsequently decodes these encodings with the target model to construct target images.

*2.2.2 Diffusion Bridge Models.* Recent advancements in diffusion bridge models have shown notable progress, with some focusing on providing theoretical guidance [1], while others have demonstrated successful applications across various tasks such as image restoration [23, 42]. In particular, BBDM [22] introduces a novel approach, uniquely modeling image-to-image translation in both image and latent space as a stochastic Brownian Bridge process. This departure from traditional conditional models opens new avenues for image translation tasks. BBDM algorithm is modified from a standard symmetric linear Brownian bridge by avoiding the large variance in the middle steps which causes the framework untrainable. The posterior distribution of the reverse process of BBDM is computed with the Bayes' rule. However, this BBDM formulation, NOT grounded in SDE theory, is complex and inflexible, and as a result, it does not provide a solid foundation for further improvement.

In this work, we propose a score-based Brownian bridge scheme based on SDEs that sets a solid theoretic ground beyond intuitive

ideas. Although this scheme is one kind of bridge models and bears the structural features of bridge models as the aforementioned ones, it possesses distinctive characteristics. Unlike the other bridge models, this new model is constructed directly from the SDE. Indeed, this new SDE perspective provides insights and options in image-to-image Brownian bridges and presents explicit solutions in closed forms. We particularly implemented a linear A-Bridge model (introduced in Section 4) and achieved record breaking performance in sample quality and diversity following the guidance of its SDE structure.

## 3 AMENABLE BROWNIAN BRIDGES

In this section, we present a general framework of the score-based image-to-image Brownian bridges in terms of SDE. Our goal is to seek a condition on a Brownian bridge SDE that renders an explicit functional equation or formula of the solution in the closed form so that it facilitates the training of the diffusion model. We call such a Brownian bridge *amenable*. Let $X(t)$, $0 \leq t \leq 1$, be an $n$-dimensional stochastic process and $W(t)$, $0 \leq t \leq 1$, an $m$-dimensional Brownian motion. Suppose $\mathbf{F} : \mathbb{R}^n \to \mathbb{R}^n$ and $\mathbf{G} : \mathbb{R}^n \times [0, 1] \to \mathbb{R}^{n \times m}$ are respectively vector or matrix-valued smooth functions with $\mathbf{F}(\mathbf{0}) = \mathbf{0}$. We consider an SDE with $X_0 \in \mathbb{R}^n$ that forms a Brownian bridge which pins $X(1) = \mathbf{0}$:

$$dX(t) = -\frac{\mathbf{F}(X(t))}{1-t} \, dt + \mathbf{G}(X(t), t) \, dW(t), \qquad (1)$$
$$X(0) = X_0$$

where $0 < t < 1$. We say the Brownian bridge expressed by the SDE (Eq. 1) is an *Amenable Brownian Bridge*, or *A-Bridge* for short, if there are smooth functions $\mathbf{u} = (u^1, \ldots, u^n) : \mathbb{R}^n \to \mathbb{R}^n$ and $M : (0, 1) \to \mathbb{R}^{n \times m}$ that satisfy the conditions

$$\mathbf{u}(X) = \mathbf{0} \ (X \in \mathbb{R}^n) \iff X = \mathbf{0}, \qquad (2)$$

$$D\mathbf{u} \cdot \mathbf{F}(X) = \mathbf{u}(X) + \frac{1-t}{2} \left[ D^2\mathbf{u} : \mathbf{G}, \mathbf{G} \right](X, t) \qquad (3)$$
$$D\mathbf{u}(X) \cdot \mathbf{G}(X, t) = M(t), \ (X \in \mathbb{R}^n, t \in (0, 1))$$

where $D\mathbf{u}$, $D^2\mathbf{u}$ are the gradient and Hessian of $\mathbf{u}$, and

$$\left[ D^2\mathbf{u} : \mathbf{G}, \mathbf{G} \right]_k (X, t) = \text{tr} \left( \mathbf{G}(X, t)^T D^2 u^k(X) \mathbf{G}(X, t) \right).$$

A functional equation can be garnered from an A-Bridge $X(t), 0 \leq t \leq 1$, as stated in Theorem 3.1 below. A proof is included in the Supplementary Materials.

THEOREM 3.1. *Suppose (1) is amenable. Then $X_t := X(t)$ verifies the functional equation*

$$\mathbf{u}(X_t) = (1-t)\mathbf{u}(X_0) + (1-t) \int_0^t \frac{M(\tau)}{1-\tau} \, dW_\tau \qquad (4)$$

*Consequently, for the solution $X(t)$ of (1), it holds that $\lim_{t \to 1^-} X(t) = \mathbf{0}$ almost surely.*

A family of A-Bridges are given when $\mathbf{F}(X) = aX$ for constants $a$ and $\mathbf{G}(X, t) = \mathbf{G}(t)$ that depends on $t$ only.

A continuous Gaussian process has a reversal in time in an identical functional form as the forward process [10], and in this case, the reverse generative process to Eq. 1 can be expressed as a Brownian bridge SDE [2]

$$dX(t) = -\frac{\tilde{\mathbf{F}}(X(t))}{1-t} \, dt + \mathbf{G}(X(t), t) \, d\overline{W}, \qquad (5)$$
$$X(1) = \mathbf{0}$$

where $0 < t < 1$. Here $\overline{W}(t)$ is a reverse-time Brownian motion driven by the stochastic process

$$d\overline{W}(t) = dW(t) + \frac{\nabla_X \cdot \left[ G^T(X(t),t) p(X(t),t) \right]}{p(X(t),t)} \, dt$$
$$\overline{W}(0) = \mathbf{0},$$

and

$$\tilde{\mathbf{F}}(X(t)) = \mathbf{F}(X(t))$$
$$+ \frac{1-t}{p(X(t),t)} \nabla_X \cdot \left( \mathbf{G}(X(t), t) \mathbf{G}(X(t), t)^T p(X(t), t) \right),$$

where $p(X(t), t)$ is the probability distribution of $X(t)$.

If we want to pin a nonzero terminal state $Y$, we just consider $X(t) - Y$ in place of $X(t)$ in an A-Bridge specified by Eq. 1. In this case, the function $\mathbf{F}(X)$ is replaced by $\mathbf{F}(X - Y)$.

In this paper, we focus on a linear A-Bridge model when $\mathbf{F}(X) = X - Y$ and $\mathbf{G}(t) = \lambda\sqrt{1 - t}I$ with a parameter $\lambda > 0$, where $I$ denotes the $n \times n$ identity matrix. Note that our SDE-based framework provides many options when selecting an A-bridge as long as the conditions given in Eqs. 2 and 3 are satisfied. In general, determining the best noise schedule in diffusion models for a given dataset is still an open problem; for example, the optimal noise schedule for the well-known DDPM model [15] is not yet clear. Here, we choose the asymmetric variance $\mathbf{G}(t) = \lambda\sqrt{1 - t}I$ on $[0, 1]$ for two reasons. The first is the intuitive idea that by diminishing the variance of the white noise near 1 the A-Bridge should converge to $Y$ fast and well as $t \to 1^-$. Second, we want to test that the A-Bridge model is robust enough that it delivers results of high quality for the non-symmetric variance. Our experimental results (in Section 5) positively confirm both.

## 4 THE LINEAR A-BRIDGE MODEL

We now establish a linear A-Bridge model for image-to-image translation that aims to achieve high image quality and sample diversity and enable efficient accelerated sampling. The mean of this model is chosen as a linear function of $t$ because of its simplicity and clarity. In the mean time, its nonsymmetric variance over $[0, 1]$, obtained directly from SDE, is one of the key differences (in addition to SDE-based vs. non-SDE-based) between our model and the existing ones [22].

### 4.1 Linear A-Bridge SDE

We construct an $n$-dimensional amenable diffusion Brownian bridge process $X(t), 0 \leq t \leq 1$, with pinned ends $X(0) = X_0$ and $X(1) = Y$. The forward inference SDE of the linear A-Bridge that we adopt is the following:

$$dX_t = -\frac{X_t - Y}{1-t} \, dt + \lambda\sqrt{1 - t} \, dW_t \qquad (6)$$
$$X(0) = X_0$$

where $0 < t < 1$, $W_t$ is an $n$-dimensional Brownian motion, and $\lambda > 0$ is a parameter. Let $p(X_t, t)$ denote the probability distribution of $X(t), 0 \leq t \leq 1$. The SDE in Eq. 6 is an amenable Brownian bridge (see the Supplementary Materials for details). The functional equation 4 in Theorem 3.1 in this case takes an explicit form

$$X_t = (1-t)X_0 + tY + \lambda(1-t) \int_0^t \frac{dW_\tau}{\sqrt{1 - \tau}} \qquad (7)$$

which gives the marginal distribution of the forward process. That is,

$$X_t \sim \mathcal{N}\left((1-t)X_0 + tY, \lambda^2(1-t)^2 \int_0^t \frac{d\tau}{1-\tau} I\right). \qquad (8)$$

In particular, $X_1 = Y$. The transitional distribution is given by, for $0 \le s < t \le 1$:

$$X_t = \frac{1-t}{1-s}X_s + \frac{t-s}{1-s}Y + \lambda(1-t)\int_s^t \frac{dW_\tau}{\sqrt{1-\tau}}, \qquad (9)$$

or that

$$X_t \sim \mathcal{N}\left(\frac{1-t}{1-s}X_s + \frac{t-s}{1-s}Y, \lambda^2(1-t)^2 \int_s^t \frac{d\tau}{1-\tau}\right).$$

A fact that is worth noting is that both marginal and transitional distributions are exact due to the amenability of the A-Bridge (Eq. 6).

The reverse Brownian bridge SDE of Eq. 6 takes the form, [10] and [2],

$$dX_t = -\left(\frac{X_t - Y}{1-t} + \lambda^2(1-t)\nabla_X \log p(X_t, t)\right) dt$$
$$\qquad\qquad + \lambda\sqrt{1-t}\, d\overline{W}_t \qquad (10)$$
$$X(1) = Y$$

where $0 < t < 1$, $\nabla_X \log p(X_t, t)$ is the score function, and $\overline{W}_t$ is an $n$-dimensional reverse-time Brownian motion. In theory, the reverse generative process furnishes one with the output $X(0) = X_0$.

The task of the A-Bridge amounts to computing the score of the marginal distribution $\nabla_X \log p(X_t, t)$ at each time step $t$ and simulating the reverse generative process to produce samples $\hat{X}_0$ of $X_0$.

## 4.2 Discretization of A-Bridge SDE

Given two image sets $\mathbb{X}$ and $\mathbb{Y}$, we sample a pair $(I, Y)$, where $I \in \mathbb{X}$ and $Y \in \mathbb{Y}$. When necessary, we put a subscript $k$ to $I$ and $Y$ to indicate we sample over all corresponding image pairs from $\mathbb{X}$ and $\mathbb{Y}$. Let $X_0 = I$ and $T$ be the number of time steps. In the forward inference process, for each $t = 1, \dots, T$, we define $X_t = \left(1 - \frac{t}{T}\right)X_0 + \frac{t}{T}Y + B(t)\,\epsilon$ according to Eq. 7, where $\epsilon \sim \mathcal{N}(0, I)$, and

$$B(t) = \begin{cases} \lambda(1-\frac{t}{T})\sqrt{\ln\left(\frac{1}{1-\frac{t}{T}}\right)} & \text{if } t = 1, \dots, T-1 \\ 0 & \text{if } t = T \end{cases}$$

As the linear A-Bridge is amenable, the formula of $X_t$ is exact instead of an approximate value at each step [22]. $B(T)$ is as defined to avoid a blowup because $\lim_{t \to T^-} B(t) = 0$. We train a neuron network to approximate the difference of $X_t$ and $X_0$, i.e. $\epsilon_\theta$ [22], instead of $\epsilon$ as in DDPM [15]:

$$\nabla_\theta \left\| \frac{t}{T}(Y - X_0) + B(t)\epsilon - \epsilon_\theta(X_t, t) \right\|^2$$

for $\epsilon \sim \mathcal{N}(0, I)$, $Y$, and $X_t$ defined as above. More precisely, we train $\epsilon_\theta$ by minimizing

$$E_{I_k, Y_k, \epsilon \sim \mathcal{N}(0,I), t \in \{1,\dots,T\}} \left\| \frac{t}{T}(Y_k - I_k) + B(t)\epsilon - \epsilon_\theta(X_t, t) \right\|^2$$

where

$$X_t = \left(1 - \frac{t}{T}\right)I_k + \frac{t}{T}Y_k + B(t)\epsilon.$$

---

**Algorithm 1** Training Algorithm for the linear A-Bridge

1: **function** TRAIN($I_A, I_B$)
2:     **repeat**
3:         $X_0 \leftarrow I_A, Y \leftarrow I_B$
4:         $t \sim \text{Uniform}(1, \dots, T)$
5:         $\epsilon \sim \mathcal{N}(0, I)$
6:         $X_t = (1 - \frac{t}{T})X_0 + \frac{t}{T}Y + B(t)\epsilon$
7:         $G \leftarrow \nabla_\theta \left\| \frac{t}{T}(Y - X_0) + B(t)\epsilon - \epsilon_\theta(X_t, t) \right\|^2$
8:         GradientStep($G$)
9:     **until** converged
10: **end function**

---

In the reverse generative process, we take $X_T = Y$. For the A-Bridge, the probability distribution $p(X_t, t)$ and hence the score function $\nabla_X \log p(X_t, t)$ can be calculated in a closed form from Eq. 7 or Eq. 8. Plugging the computed value of the score function into the reverse SDE (10), we can discretize Eq. 10 to get the discrete reverse transitional formula for $t = T, \dots, 2$

$$X_{t-1} = C_{xt} X_t - C_{yt} Y - C_{\epsilon t} \epsilon_\theta(X_t, t) - C_{zt} z,$$

where $z \sim \mathcal{N}(0, I)$, and

$$C_{xt} = \frac{1}{C}\left(1 + \frac{1}{T \ln \frac{T}{T-t+1}}\right)$$

$$C_{yt} = \frac{1}{C}\left(\frac{1}{T-t+1} - \frac{t-1}{T(T-t+1)\ln\frac{T}{T-t+1}}\right)$$

$$C_{\epsilon t} = \frac{1}{C}\frac{1}{T \ln\frac{T}{T-t+1}}$$

$$C_{zt} = \frac{\lambda}{C}\sqrt{1 - \frac{t}{T} + \frac{1}{T}}\sqrt{\frac{1}{T}}$$

$$C = 1 - \frac{1}{T-t+1} + \frac{1}{(T-t+1)\ln\frac{T}{T-t+1}}.$$

Finally, we garner a sample of $X_0$ by setting

$$X_0 = X_1 - \epsilon_\theta(X_1, 1).$$

The reverse process generates a sample $\hat{I}$ of the image $I$. In the discretization of the A-Bridge, we take care to avoid the blowup of the coefficients when $t = T$. In fact, the blowup is avoided through the definition of $B(t)$ in the forward formula and the choice of $\frac{t-1}{T}$ in the discretizaion of the reverse formula (Eq. 10). In addition, $X_0$ in the reverse formula at the intermediate steps is replaced by $X_t - \epsilon_\theta(X_t, t)$ as suggested in [15] and [22]. The discrete forward and reverse processes are detailed in Algorithms 1 and 2. We emphasize that we discretize the forward and reverse SDE according to the flow of time. That is, in the forward training, we discretize the SDE at time $t$, while the SDE is discretized at time $t - \Delta t$ in the sampling.

## 4.3 Training Objective

Instead of optimizing the negative log likelihood function $L = -E_{q(X_0)} \log p(X_0)$ directly, we set the training objective as minimizing the Evidence Lower Bound (ELBO) of $L$ [32]:

$$K = \sum_2^T D_{KL}\left(q(X_{t-1}|X_t, X_0, Y) | p_\theta(X_{t-1}|X_t, Y)\right)$$

---

**Algorithm 2** Sampling Algorithm for the linear A-Bridge

---

1: **function** SAMPLE($Y$)
2:      $X_T \leftarrow Y$
3:      **for** $t = T, \ldots, 2$ **do**
4:          $z \sim \mathcal{N}(0, I)$
5:          $X_{t-1} \leftarrow C_{xt} X_t - C_{yt} Y - C_{\epsilon t} \epsilon_\theta(X_t, t) - C_{zt} z$
6:      **end for**
7:      $X_0 \leftarrow X_1 - \epsilon_\theta(X_1, 1)$
8:      **return** $X_0$
9: **end function**

---

where $D_{KL}(q|p) = E_q\left[\log \frac{q(x)}{p(x)}\right]$ is the *Kullback-Leibler divergence* of $q$ relative to $p$, and the posterior distribution

$$q(X_{t-1}|X_t, X_0, Y) = \frac{q(X_t|X_{t-1}, Y)q(X_{t-1}|X_0, Y)}{q(X_t|X_0, Y)}$$

is determined by the Bayes' rule. The first term in the KL divergences has been dropped as it is a constant. Here $q(X_{0:T})$ and $p(X_{0:T})$ denote respectively the forward and reverse trajectories of the A-Bridge.

## 4.4 Accelerated Sampling

The reverse sampling process of our linear A-bridge model can be accelerated [33] in line with the forward non-Markovian inference process defined below. Let $\tau = \{\tau_1, \tau_2, \ldots, \tau_S\}$ be a subsequence of $\{1, \ldots, T\}$ with $\tau_S = T$ and $\sigma = (\sigma_1, \ldots, \sigma_T) \in \mathbb{R}_+^T$ be a real vector. Set $m_t = \frac{t}{T}$.

We recall that, for all $t = 1, \ldots, T$,

$$q(X_t|X_0, Y) = \mathcal{N}\left((1 - m_t)X_0 + m_t Y, B^2(t) I\right).$$

We can define

$$q(X_{t-1}|X_t, X_0, Y) = \mathcal{N}((1 - m_{t-1})X_0 + m_{t-1}Y$$
$$+ \frac{\sqrt{B^2(t-1) - \sigma_t^2}}{B(t)}(X_t - (1 - m_t)X_0 - m_t Y), \sigma_t^2 I).$$

In particular, the discrete non-Markovian forward inference process (i.e., the ground truth) is defined by, for $i = 1, \ldots, S$,

$$q(X_{\tau_{i-1}}|X_{\tau_i}, X_0, Y) = \mathcal{N}((1 - m_{\tau_{i-1}})X_0 + m_{\tau_{i-1}}Y$$
$$+ \frac{\sqrt{B^2(\tau_{i-1}) - \sigma_{\tau_i}^2}}{B(\tau_i)}(X_{\tau_i} - (1 - m_{\tau_i})X_0 - m_{\tau_i}Y), \sigma_{\tau_i}^2 I).$$

The Bayes' rule reads

$$q(X_{\tau_i}|X_{\tau_{i-1}}, X_0, Y) = \frac{q(X_{\tau_{i-1}}|X_{\tau_i}, X_0, Y)q(X_{\tau_i}|X_0, Y)}{q(X_{\tau_{i-1}}|X_0, Y)}$$

from which the discrete trajectory matches $q(X_{\tau_i}|X_0, Y)$. We take $\sigma_{\tau_i} = 0.5B(\tau_{i-1})$ for every $i = S, \ldots, 2$ to guarantee the square root in the formula makes sense.

Hence, we define the reverse accelerated sampling by

$$X_{\tau_{S-1}} = \left(1 - \frac{\tau_{S-1}}{T}\right)(Y - \epsilon_\theta(Y, T)) + \frac{\tau_{S-1}}{T}Y + \sigma_S z,$$

$$X_{\tau_{i-1}} = \tilde{Y}_{\tau_{i-1}} + \frac{\sqrt{B^2(\tau_{i-1}) - \sigma_{\tau_i}^2}}{B(\tau_i)}(X_{\tau_i} - Y_{\tau_i}) + \sigma_{\tau_i}z,$$

for $i = S - 1, S - 2, \ldots, 2$, where $z \sim N(0, I)$,

$$\tilde{Y}_{\tau_{i-1}} = (1 - m_{\tau_{i-1}})\left(X_{\tau_i} - \epsilon_\theta(X_{\tau_i}, \tau_i)\right) + m_{\tau_{i-1}}Y,$$
$$Y_{\tau_i} = (1 - m_{\tau_i})\left(X_{\tau_i} - \epsilon_\theta(X_{\tau_i}, \tau_i)\right) + m_{\tau_i}Y,$$

and lastly,

$$X_0 = X_{\tau_1} - \epsilon_\theta(X_{\tau_1}, \tau_1).$$

Algorithm 3 provides details of the accelerated sampling. Note that one can opt to allocate a larger number of sampling steps to the early phase of the reverse process as the generation tendency becomes more pronounced in the initial steps of sampling [24].

---

**Algorithm 3** Accelerated Sampling for the Linear A-Bridge

---

1: **function** FASTSAMPLE($Y$)
2:      $\tau_S \leftarrow T$
3:      $X_{\tau_S} \leftarrow Y$
4:      $X_{\tau_{S-1}} \leftarrow \left(1 - \frac{\tau_{S-1}}{\tau_S}\right)(X_{\tau_S} - \epsilon_\theta(X_{\tau_S}, \tau_S)) + \frac{\tau_{S-1}}{\tau_S}X_{\tau_S} + \sigma_S z$
5:      **for** $i = S - 1, \ldots, 2$ **do**
6:          $z \sim \mathcal{N}(0, I)$
7:          $X_{\tau_{i-1}} \leftarrow \tilde{Y}_{\tau_{i-1}} + \frac{\sqrt{B^2(\tau_{i-1}) - \sigma_{\tau_i}^2}}{B(\tau_i)}(X_{\tau_i} - Y_{\tau_i}) + \sigma_{\tau_i}z$
8:      **end for**
9:      $X_0 \leftarrow X_{\tau_1} - \epsilon_\theta(X_{\tau_1}, \tau_1)$
10:      **return** $X_0$
11: **end function**

---

## 5 EXPERIMENTS

### 5.1 Implementation details

To ensure a fair comparison with current methods in the literature, our approach significantly draws upon existing coding framework. We incorporated identical key components, notably the UNet architecture and the same pretrained VQGAN model [9] that used in both BBDM [22] and Latent Diffusion Model [30], to maintain consistency in experimental setup. In the training phase, we configured the number of time steps ($T$) for the linear A-Bridge model to be 1,000. During the inference phase, we implemented the fast sampling approach (Algorithm 3) with 200 steps. This strategy was designed to enhance computational efficiency while ensuring the quality of the samples. Following [22], all of our models were trained for 100 epochs using Adam optimizer with a learning rate of 0.0001.

All experiments were implemented using PyTorch [29] and ran on a machine with two NVIDIA RTX A6000 GPUs. Our source code is available in the supplementary materials.

### 5.2 Experiment Setup

**Datasets and Tasks:** In this work, we focused on evaluating several key tasks in the field of image-to-image translation. We selected five datasets, each tailored to evaluate three different tasks. The specifics of these datasets and their associated tasks are detailed as follows:

- Semantic synthesis task: Evaluated on the CelebA-Mask-HQ dataset [19] and the Cityscapes dataset [7].
- Edges to photos task: Evaluated on the Edges2Shoes and Edges2Handbags datasets [17].

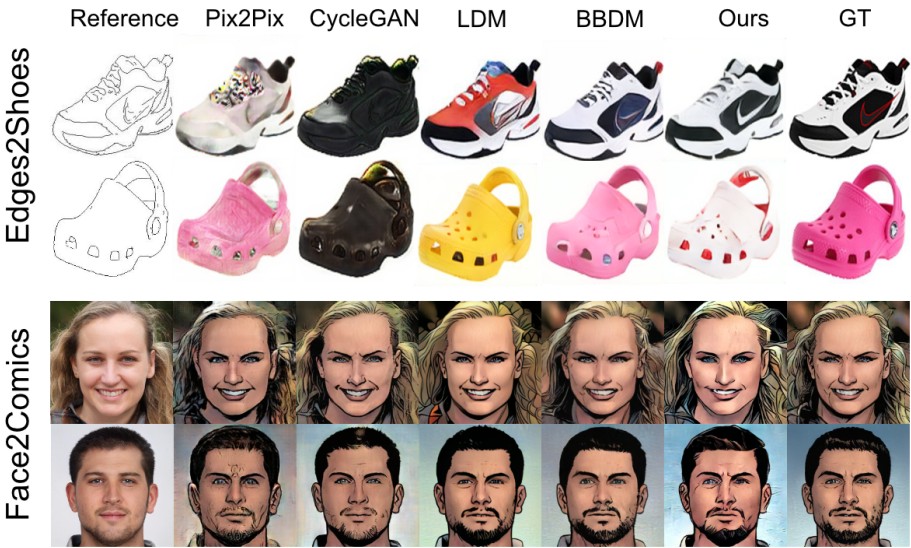

**Figure 2: The qualitative comparison between our linear A-Bridge and other methods on Edges2Shoes and Face2Comics datasets. GT stands for Ground Truth.**

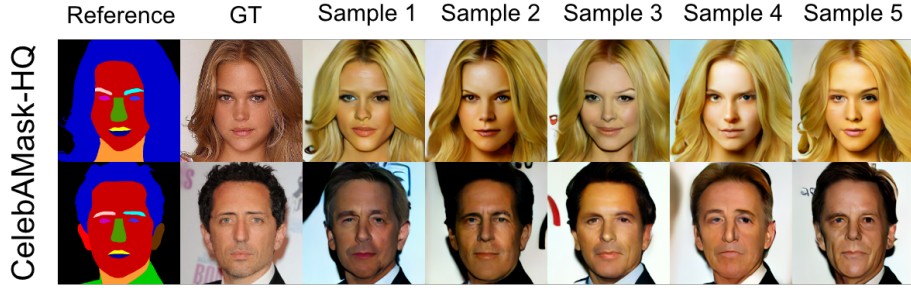

**Figure 3: The diversity samples of our linear A-Bridge on the CelebAMask-HQ dataset**

- Style transfer task: Evaluated on the Faces2Comics dataset [37]

Following the experimental setup of existing methods [22, 42], we resized Cityscapes and CelebAMask-HQ images to $256 \times 256$, Edges2Handbags images to $64 \times 64$, while keeping the original resolutions for the other datasets. Following the literature, we conducted our experiments in the latent space for all the datasets except Edges2Handbags, which was done in the image space.

**Metrics:** In our evaluation, we employed the widely used Fréchet Inception Distance (FID) [14] to assess the quality of the generated images. Additionally, we measured the diversity of the samples using the Learned Perceptual Image Patch Similarity (LPIPS) [41] metric. Following literature, we replaced the LPIPS metric with the mean Intersection over Union (mIoU) using a pre-trained segmentation network developed in [17] for our evaluation on the Cityscapes dataset.

**Baseline:** While our primary focus is on comparing our method to SOTA of image-to-image translation (e.g, BBDM in [22]), we have also expanded our analysis to include a large variety of baselines.

These baselines encompass Pix2Pix [17], CycleGAN [43], DRIT++ [21], CDE [31], and LDM [30]. Notably, Pix2Pix, CycleGAN, and DRIT++ are based on conditional GANs, while CDE and LDM utilize conditional diffusion models for image translation. Additionally, we also compared with other recent models from the bridge family, such as I2SB [23] and DDBM [42]. It is worth noting that I2SB indicates their model is primarily focused on addressing image restoration tasks, and the majority of DDBM experiments were conducted in the image space. So, their experimental setting is different from other models. To make a fair comparison, we compare with them separately from other models and conduct experiments on the Edges2Handbags dataset using their established setting.

In addition, for a comprehensive evaluation on the CelebAMask-HQ dataset, we have included comparisons with OASIS [36] and SPADE [28]. Similarly, for the Cityscapes dataset, we have benchmarked our method against recent SOTA techniques including CUT, FastCUT [27], CycleGAN, MUNIT [16], DRIT [20], DistanceGAN [4], SelfDistance [4], GCGAN [11], DCLGAN [13], and SimDCL [13].

| Model | CelebAMask-HQ | | edges2shoes | | faces2comics | |
|---|---|---|---|---|---|---|
| | FID ↓ | LPIPS ↑ | FID ↓ | LPIPS ↑ | FID ↓ | LPIPS ↑ |
| Pix2Pix | 56.997 | 0.431 | 36.339 | 0.183 | 49.964 | 0.282 |
| CycleGAN | 78.234 | **0.490** | 66.115 | 0.276 | 35.133 | 0.263 |
| DRIT++ | 77.794 | 0.431 | 53.373 | **0.498** | 28.875 | 0.285 |
| SPADE | 44.171 | 0.376 | - | - | - | - |
| OASIS | 27.751 | 0.384 | - | - | - | - |
| CDE | 44.171 | 0.376 | 21.189 | 0.196 | 33.983 | 0.259 |
| LDM | 22.816 | 0.371 | 13.020 | 0.173 | 24.280 | 0.205 |
| BBDM | 21.350 | 0.370 | 10.924 | 0.183 | 23.203 | 0.192 |
| **Linear A-Bridge (Ours)** | **12.832** | 0.482 | **9.457** | 0.192 | **12.711** | **0.368** |

**Table 1: Quantitative comparison on CelebAMask-HQ, edges2shoes, and faces2comics datasets. A - indicates that the metric was not reported by the method, the same applied in Table 2.**

| Method | FID ↓ | mIoU ↑ |
|---|---|---|
| Pix2Pix | - | 0.17 |
| CycleGAN | 68.6 | - |
| CRN | - | 0.20 |
| MUNIT | 91.4 | - |
| DRIT | 155.3 | - |
| Distance | 85.8 | - |
| SelfDistance | 78.8 | - |
| CGGAN | 105.2 | - |
| CUT | 56.4 | - |
| FastCUT | 68.8 | - |
| DCLGAN | 49.4 | 0.17 |
| DRGAN | - | 0.19 |
| SimDCL | 51.3 | - |
| **Linear A-Bridge (ours)** | **32.9** | **0.23** |

**Table 2: Comparison of FID and mIoU scores across different methods on the Cityscapes dataset with the best scores bolded.**

This diverse set of baselines ensures a comprehensive and robust comparative evaluation.

## 5.3 Performance Comparison

*5.3.1 Qualitative comparison:* In this section, we qualitatively compare our proposed linear A-bridge model with other image-to-image translation models. We categorize these models into three main classes and select representative models for comparison. The first category includes GAN-based models, for which we have chosen well-known models pix2pix and CycleGAN. The second category consists of conditional diffusion models, and we have chosen Latent Diffusion Model (LDM) as a representative. The third category comprises bridge models that directly let the model to learn two distributions, e.g., BBDM.

As shown in Figure 2, GAN-based models often struggle to generate intuitively good images. In contrast, the conditional diffusion model - LDM can generate high-quality images in most cases, but sometimes with minor imperfections. For instance, in the second row of the edges2shoes examples in Figure 2, the colors in the shoe holes may not precisely match the inner colors. Bridge-type models

in general produce more realistic and higher quality images. Our linear A-bridge, in particular, derives precise solutions from the perspective of SDE, allowing our model to recognize subtle details that may go unnoticed by other models. For example, in the first example of edges2shoes in Figure 2, only the linear A-bridge model correctly identifies the logo on the reference in the generated image.

The images generated by our model not only excel in image quality but also exhibit strong diversity. As depicted in Figure 3, our model demonstrates competitive diversity, both at a macro (e.g., image colors, backgrounds) and a micro level (e.g., facial expressions and patterns), while maintaining high image quality. More comparison and diversity examples are provided in the supplemental materials.

*5.3.2 Quantitative comparison:* As shown in Table 1, our linear A-Bridge model represents a significant advancement in the area of image-to-image translation, particularly noted by its impressive reductions in the FID score over current SOTA. For example, on the CelebAMask-HQ dataset, the linear A-Bridge delivers an FID of 12.832, marking a 40% improvement compared to the second best (the BBDM model), which scores 21.350. This major improvement in FID underscores our model's capability to generate images that are much closer to the real image distribution, essential for applications that require high-quality image translations.

The exceptional performance of our model extends to the edges2-shoes and faces2comics datasets. For edges2shoes, the linear A-Bridge shows a 13% improvement in FID, while for faces2comics, the enhancement is a striking 45% over BBDM (the second best). As indicated by the LPIPS scores, these improvements are especially noteworthy given that the linear A-Bridge also maintains a higher level of diversity than BBDM. Furthermore, as shown in Table 2, our linear A-Bridge also achieved a notable improvement in FID on the Cityscapes dataset. We get 32.9 while the second best is 51.3 (by SimDCL).

Among models with similar LPIPS scores, our model markedly outperforms in terms of FID. As demonstrated in Table 1, the linear A-Bridge achieved the second best LPIPS (0.482) among all models on CelebAMask-HQ dataset. Though CycleGAN exhibited a slightly higher diversity with a LPIPS of 0.490, its FID is 6.5 times worse (higher) than our model (78.234 vs. 12.832). Similarly, our model closely matches the LPIPS of CDE: 0.196 (CDE) vs. 0.192 (ours) on

| Method | FID ↓ |
|---|---|
| I2SB | 7.43 |
| DDBM (VE) | 2.93 |
| DDBM (VP) | 1.83 |
| **Linear A-Bridge (Ours)** | **1.07** |

Table 3: Comparison of FID across different diffusion bridge methods on the Edge2Handbags dataset with the best scores bolded.

| $\lambda$ | FID ↓ | LPIPS ↑ |
|---|---|---|
| 1 | 12.939 | 0.204 |
| 2 | **9.457** | 0.192 |
| 3 | 13.2286 | **0.235** |

Table 4: Ablation study for different $\lambda$ values on the edges2shoes dataset with the best scores bolded.

the edges2shoes dataset, yet surpasses it on FID by over 55%: 21.189 (CDE) vs. 9.457 (ours). This highlights the exceptional ability of our linear A-Bridge to significantly enhance image quality while preserving ample diversity.

Finally, as shown in Table 2, the linear A-Bridge is able to exhibits its versatility by scoring an mIoU of 0.23 on Cityscapes dataset, not for segmentation accuracy but as a novel metric in this context, to assess the consistency and accuracy of attribute translation in synthesized images. This score is 15% higher than that of the second best (the CRN model), which serves as further evidence of its superior performance in preserving and replicating image details.

*5.3.3  Compare with other diffusion bridge models:* Table 3 presents a comparison between our linear A-Bridge model and SOTA diffusion bridge models within the image space on the edge2handbags dataset. Our model demonstrated a 44% enhancement in the FID, registering an impressive 1.07, whereas the next closest competitor, the VP DDBM, scored 1.83. Remarkably, we achieved these results through fast sampling by utilizing just 3 time steps. This underscores the linear A-Bridge model's ability to deliver high-quality outcomes with minimal time steps, compared to I2SB and DDBM, which utilized 40 time steps. Our model's superior performance is evident. Figure 4 displays some sampling results of our model on the Edges2Handbags dataset.

## 5.4  Ablation study

*5.4.1  Impact of $\lambda$.* : our linear A-Bridge contains a hyper-parameter $\lambda$, which controls the variance of $X_t$ at each step of the Brownian bridge. An ablation study was conducted, applying varying values of $\lambda$ within our model on the edges2shoes dataset to investigate its impact on image quality and diversity.

Our result in Table 4 indicates that the choice of $\lambda$ critically governs the balance between the fidelity and diversity of the generated images. A certain value of $\lambda$ was observed to yield the highest quality in generated images, suggesting an optimal alignment with the target image distribution. However, this peak in quality was associated with a slight reduction in diversity, implying a potential limitation on the variation within the generated images. Conversely,

| Sampling Steps | Algorithm | FID ↓ | LPIPS ↑ |
|---|---|---|---|
| 50 | Alg. 3 | 15.25 | 0.417 |
| 100 | Alg. 3 | 13.00 | 0.431 |
| 200 | Alg. 3 | 12.83 | **0.482** |
| 1000 | Alg. 2 | **9.89** | 0.421 |

Table 5: Quantitative scores with different numbers of sampling steps on CelebAMask-HQ. The algorithm marked as "Alg.2" represents regular sampling, while "Alg.3" denotes fast sampling.

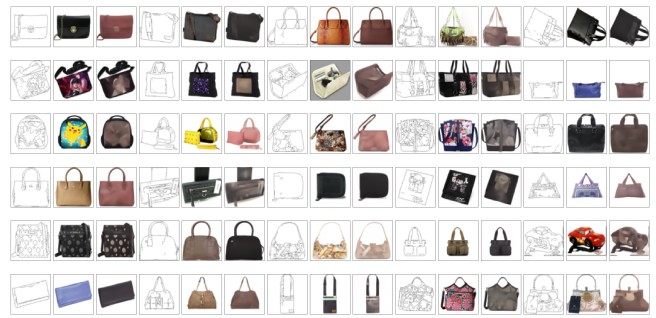

Figure 4: Random selected qualitative sampling results on Edges2Handbags by our linear A-bridge. Each row of results consists of three images: the first is the input, the second is the ground truth, and the third is the sampled result.

other values of $\lambda$ were found to favor diversity, which could be beneficial for scenarios that demand a wide array of image outputs. Nevertheless, this diversity comes at the cost of diminishing certain quality aspects of the images. Based on this ablation study, we choose a $\lambda$ value of 2 in all our experiments.

*5.4.2  Impact of sampling steps.* : To investigate the impact of sampling steps on model performance, we employed four different sampling steps and did an ablation on CelebAMask-HQ. As shown in Table 5, we noticed that, in general, the quality (FID) of the generated images decreases with a less number of sampling steps. To strike a good balance between sampling quality and diversity, and also between sampling quality and time, we employed 200 as the default sampling step for all our experiments.

## 6  CONCLUSION AND FUTURE WORK

We presented a framework for image-to-image translations based on A-Bridges. Introduction of the A-Bridge models opens the gate for new sampling algorithms in line with this new understanding of amenable SDE. Following the guidance of its SDE structure, our linear A-Bridge model demonstrated its effectiveness in significantly improving image quality compared to SOTA algorithms while maintaining ample diversity. Its flexibility also provides the opportunity to keep balance between the image quality and diversity on a particular image set. Meanwhile, the answer to questions like how to choose the best A-Bridge model (e.g, the best noise scheduling of the Brownian bridge model) to run on a specific image set remains largely open and will be our future work.

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
