# OpenReview forum: "Score-Based Image-to-Image Brownian Bridge"
_acmmm.org/ACMMM/2024/Conference — MM2024 Poster_

### Official Review · Reviewer_QYKi · 2024-05-08

**Rating:** 4
**Confidence:** 1

**Summary:**

The author proposes a score-based Stochastic Differential Equation (SDE) approach via the Brownian bridges, termed the Amenable Brownian Bridges (A-Bridges), to image-to-image translation tasks as an unconditional diffusion model.

**Strengths:**

Extensive experiments on various benchmark datasets, proposed linear A-Bridge model, following the guidance of its SDE structure, exhibits a significant enhancement over current SOTA for image-to-image translation tasks.

**Limitations:**

(1) The method proposed by the author requires the use of paired data sets, which limits the practical use of the method.

(2) It is suggested to add comparative experiments, including VCT [1].

[1] Cheng B, Liu Z, Peng Y, et al. General image-to-image translation with one-shot image guidance[C]//Proceedings of the IEEE/CVF International Conference on Computer Vision. 2023: 22736-22746.

**Suitability:**

2

---

### Official Review · Reviewer_vduf · 2024-05-21

**Rating:** 4
**Confidence:** 3

**Summary:**

This paper proposes an image-to-image translation approach leveraging fractional-based stochastic differential equations (SDEs) within an unconditional diffusion model framework, termed Amenable Brownian Bridges. This framework comprises a large family of Brownian Bridge models, provides the explicit solution of in a closed form and thus facilitates model training. The authors also verified that the proposed method could accelerate sampling process guided by its SDE structure.

**Strengths:**

1. The Amenable Brownian Bridges SDE simplifies both the forward and reverse diffusion processes unconditionally, thereby improving the understanding and implementation of image-to-image mapping. This approach integrates a wide range of Brownian Bridges, providing a clear and explicit closed-form solution. Notably, among the numerous potential Bridges, they opt for the discretized linear A-bridge model and introduce an accelerated sampling algorithm.
2. The proposed A-Bridge model demonstrates improvements over the current state-of-the-art (SOTA) methods for image-to-image translation tasks.

**Limitations:**

1. Please clarify and explain the superiority of linear A-bridge model compared to the art. From the qualitative results, it appears that the linear A-Bridge proposed in this paper does not exhibit significant advantages in the Edges2Shoes and Face2Comics datasets. While the linear A-Bridge streamline the unconditional forward and reverse diffusion processes, offering a more flexible translation approach, the experiments presented in this paper may not provide a comprehensive test for this.
2. From a quantitative analysis perspective, the method proposed in this paper has shown improvements in image quality indicators such as FID, but has seen a decrease in consistency indicators such as LPIPS. The authors should provide a more detailed explanation of this issue.

**Suitability:**

3

---

### Official Review · Reviewer_wVVd · 2024-06-02

**Rating:** 3
**Confidence:** 3

**Summary:**

The authors propose a fraction-based stochastic differential equation (SDE) approach to the image-to-image translation task as an unconditional diffusion model via a Amenable Brownian Bridges.

**Strengths:**

This is a well-written paper that implements image translation using the theory of Brownian bridges.

**Limitations:**

But I have the following concerns:
1. most of the novelty of this paper is based on BBDM: Image-to-Image Translation with Brownian Bridge Diffusion Models, and the real contribution needs to be further explained.
2. the comparison of experiments is insufficient and lacks comparison with ControlNet. As far as I know, some methods such as ControlNet have better visualization than this paper.

**Suitability:**

3

---

### Meta-Review · Area_Chair_VrzM · 2024-06-27

**Recommendation:** Accept (Poster)
**Confidence:** 3

**Metareview:**

The paper initially received 2 Borderline Accept and 1 Borderline Reject ratings. After the rebuttal, the two Borderline Accept ratings were kept as the reviewers found their concerns being addressed. The reviewer who's on the negative side failed to provide the final rating. The AC checked the paper, reviews, and rebuttal, and decided that this paper should be accepted considering the novelty of the A-Bridge model and its good performance in image-to-image translation tasks demonstrated in the paper.